# Measurements of the Neutron Lifetime

**F. E. Wietfeldt**

Department of Physics and Engineering Physics, Tulane University, New Orleans, LA 70118, USA;
few@tulane.edu

**Abstract:** Free neutron decay is a fundamental process in particle and nuclear physics. It is the prototype for nuclear beta decay and other semileptonic weak particle decays. Neutron decay played a key role in the formation of light elements in the early universe. The precise value of the neutron mean lifetime, about 15 min, has been the subject of many experiments over the past 70 years. The two main experimental methods, the beam method and the ultracold neutron storage method, give average values of the neutron lifetime that currently differ by 8.7 s (4 standard deviations), a serious discrepancy. The physics of neutron decay, implications of the neutron lifetime, previous and recent experimental measurements, and prospects for the future are reviewed.

**Keywords:** neutron decay; neutron lifetime; beta decay

## 1. Introduction

The neutron is a basic constituent of ordinary matter; the majority of the Earth's mass is contributed by neutrons. But when freed from the confines of a stable atomic nucleus the neutron is unstable. It decays via the weak nuclear force into a proton, an electron, and antineutrino, with a mean lifetime of about fifteen minutes. Free neutron decay is the prototype for nuclear beta decay and other semileptonic weak particle decays. The experimental value of the neutron mean lifetime gives the probability of neutron decay and determines the strength of other charged weak processes involving free protons and neutrons that are important in astrophysics, cosmology, solar physics, and neutrino detection. The neutron lifetime is a key parameter in Big Bang nucleosynthesis; its value determines the theoretical primordial helium abundance. It is an essential ingredient in the unitarity test of of the Cabibbo-Kobayashi-Maskawa (CKM) quark mixing matrix using neutron decay. The importance of the neutron lifetime has been long recognized, it has been the subject of more than 20 major experiments starting with Robson's first measurement in 1951 [1].

## 2. Theoretical Discussion

Nuclear beta decay is the physical transformation of a neutron into a proton or vice versa inside an atomic nucleus to produce the nucleus of a neighboring element in the periodic table. At the heart of beta decay is the charged-current weak interaction that enables up and down quarks to exchange identity. Nuclear beta decay will occur whenever it is energetically favored, i.e., when it results in a net decrease in total mass from the parent to the daughter system. For example, in the $A$ = 14 system $^{14}$C decays by the transformation of a neutron to a proton ($\beta^-$ decay) into the lower mass $^{14}$N. Similarly $^{14}$O decays by the transformation of a proton to a neutron ($\beta^+$ decay) into $^{14}$N. However $^{14}$N lies at the bottom of the $A$ = 14 mass multiplet so it is stable. The free neutron decays into a proton, electron, and antineutrino:

$$n \rightarrow p + e^- + \bar{\nu} \tag{1}$$

because there is a net decrease in total mass. The rest energy difference $(m_n - m_p - m_e)c^2 = 782$ keV is distributed as kinetic energy to the final state particles, so energy is conserved. Note that the

antineutrino mass is negligible here. The electron and antineutrino are needed to conserve momentum, electric charge, and lepton number.

The Hamiltonian for neutron decay can be written

$$\mathcal{M} = \left[ G_V \bar{p}\, \gamma_\mu n - G_A \bar{p}\, \gamma_5 \gamma_\mu n \right] \left[ \bar{e}\, \gamma_\mu \left( 1 + \gamma_5 \right) \nu \right], \tag{2}$$

where $\bar{p}$, $n$, $\bar{e}$, and $\nu$ are the relativistic spin wave functions of the proton, neutron, electron, and antineutrino. This Hamiltonian includes both vector ($G_V$) and axial vector ($G_A$) couplings so it violates parity symmetry, a notable and unique feature of the weak interaction.

From the Hamiltonian, Equation (2), one can compute the neutron decay probability per unit time using Fermi's golden rule:

$$dW = (2\pi)^{-5} \delta(E_e + E_\nu - \Delta) \frac{1}{2E_e} \frac{1}{2E_\nu} d^3 \boldsymbol{p_e} \, d^3 \boldsymbol{p_\nu} |\mathcal{M}|^2. \tag{3}$$

Here $E_e$, $\boldsymbol{p_e}$, $E_\nu$, and $\boldsymbol{p_\nu}$ are the electron and antineutrino total energy and momentum and $\Delta$ is the neutron–proton mass difference: $\Delta = 1.29333205(51)$ MeV [2]. Integration of Equation (3) over the antineutrino and electron momenta gives the beta electron energy spectrum:

$$\frac{dW}{dE_e} = \frac{G_V^2 + 3G_A^2}{2\pi^3} E_e |\boldsymbol{p_e}| \left( \Delta - E_e \right)^2, \tag{4}$$

An additional integration over electron energy gives the exponential decay rate constant:

$$W = \frac{(G_V^2 + 3G_A^2)}{2\pi^3} f_R. \tag{5}$$

Here $f_R$ is the value of the integral over the Fermi energy spectrum, including Coulomb, recoil order, and radiative corrections. The neutron lifetime $\tau_n$ is the inverse of $W$:

$$\tau_n = \frac{2\pi^3}{(G_V^2 + 3G_A^2) f_R} \tag{6}$$

in natural units, or writing the physical constants explicitly:

$$\tau_n = \frac{2\pi^3 \hbar^7}{(G_V^2 + 3G_A^2) m_e^5 \, c^4 f_R}. \tag{7}$$

Recoil order ($E_{\max}/M_n \approx 8 \times 10^{-4}$) effects include induced hadronic currents, in particular weak magnetism which, as was famously shown using the CVC hypothesis, can be computed from the neutron and proton magnetic dipole moments [3,4]. Radiative corrections are traditionally separated into the *outer* and *inner* corrections. The outer corrections, about 1.5% of the neutron lifetime, are long range electromagnetic corrections for both real (bremsstrahlung) and virtual photons, including the infrared divergence and Coloumb corrections to the electron wave function. The inner corrections, about 2%, are model-dependent short range electroweak corrections. Including updated values for these corrections, Marciano and Sirlin [5] find

$$\tau_n = \frac{G_F^2}{G_V^2 + 3G_A^2} 4908.7(1.9)\,\text{s}. \tag{8}$$

where $G_F = 1.1663787(6) \times 10^{-5}$ GeV$^{-2}$ [2] is the Fermi weak coupling constant calculated from the muon decay lifetime.

Conservation of vector current (CVC) implies that the vector weak coupling in the nucleon system has the same strength as for a bare quark, i.e., $G_V = G_F V_{ud}$, where $V_{ud}$ is the first element of the

Cabibbo-Kobayashi-Maskawa (CKM) quark mixing matrix. Axial current is not conserved so the value of $G_A$ is altered by the strong interaction in the hadronic environment. Thus $G_A = G_F V_{ud} \lambda$, where $\lambda$ is measured experimentally from neutron decay. A very important low energy test of the Standard Model is the unitarity of the first row of the CKM matrix:

$$|V_{ud}|^2 + |V_{us}|^2 + |V_{ub}|^2 = 1. \tag{9}$$

$|V_{ub}|^2$ is negligibly small so in practice this is a precise comparison of $V_{ud}$ and $V_{us}$. A true violation of this unitarity condition would be a clear sign of new physics beyond the Standard Model (BSM) at the low energy, precision frontier. For example, in a recent paper Bauman, et al. show that SUSY loop corrections could cause a departure from Equation (9) at the few $10^{-4}$ level and reveal BSM physics that lies beyond the present constraints of the Large Hadron Collider [6]. Currently the most precise determinations of both $G_V$ and $V_{ud}$ come from the $\mathcal{F}t$ values of 14 superallowed $0^+ \to 0^+$ beta decay systems yielding $V_{ud} = 0.97417(21)$ [7], a precision of $2 \times 10^{-4}$, limited by theoretical uncertainties in the radiative, isospin breaking, and nuclear structure corrections. Combining this with the 2018 Particle Data Group (PDG18) [8] recommended value of $V_{us} = 0.2243(5)$ one obtains $|V_{ud}|^2 + |V_{us}|^2 = 0.99932(47)$, in reasonable accord with unitarity. We note that a new calculation of the universal radiative correction to beta decay, $\Delta_R$, was recently presented [9]. If it stands, $V_{ud}$ from superallowed beta decay will reduce by about 0.05% and result in a CKM unitarity problem.

The weak coupling constants $G_A$ and $G_V$ govern other important charged weak interactions between free neutrons and protons, for example [10]:

$n + e^+ \leftrightarrow p + \overline{\nu}$   (Big-Bang nucleosynthesis)
$p + e^- \leftrightarrow n + \nu$   (Big-Bang nucleosynthesis, neutron star formation)
$p + p \to D + e^+ + \nu$   (solar fusion)
$p + p + e^- \to D + \nu$   (solar fusion)
$\nu + n \to e^- + p$   (neutrino detection)
$\overline{\nu} + p \to e^+ + n$   (antineutrino detection).

The neutron lifetime, along with neutron decay angular correlation measurements, can provide independent and precise values of $G_A$ and $G_V$ and furthermore place limits on physics beyond the Standard Model. New physics at a high mass scale such as leptoquarks or supersymmetry can cause effective weak scalar and tensor interactions at low energy and manifest as small departures from Standard Model predictions for low energy processes like beta decay. Limits on scalar and tensor weak currents, and hypothetical right-handed weak currents, can be obtained from a combined fit to experimental neutron decay and other beta decay observables [11,12].

## 3. The Neutron Lifetime and Big-Bang Nucleosynthesis

Protons and neutrons condensed from the quark-gluon plasma when the universe was about 1 millisecond old. They remained in thermal equilibrium via semileptonic weak interactions until the universe expanded to the point where the lepton density and temperature were too low to maintain equilibrium. This nucleon "freeze out" occured at $t \approx 1$ s. At that moment the ratio of neutrons to protons was fixed by a Boltzmann factor: $n/p = \exp(-\Delta m/kT_{\text{freeze}}) \approx \frac{1}{6}$. An additional three minutes of cooling and expansion followed before protons and neutrons were able to fuse into nuclei to form isotopes of the lightest elements: hydrogen, helium, lithium, and beryllium. This process of primordial nucleosynthesis was complete at about $t \approx 5$ min at which time virtually all neutrons were bound into $^4$He nuclei, with trace amounts in other light isotopes.

The neutron lifetime directly provides the combination $G_V^2 + 3G_A^2$ that determines the semileptonic weak interaction rate and hence $T_{\text{freeze}}$, the temperature of the universe at "freeze out". It also gives the fraction of neutrons that decayed or were removed by lepton capture prior to the completion of light element nucleosynthesis. The neutron lifetime experimental uncertainty is therefore primarily responsible for the theoretical uncertainty in the primordial helium abundance $Y_P$ [13]. Due to

its particular sensitivity to $T_{\text{freeze}}$ and the expansion rate of the early universe, $Y_P$ can be used to constrain the number of effective neutrino species $N_{\text{eff}}$ (including actual neutrinos and other light particles that may have been present). A recent analysis by Nollett and Steigman gives $N_{\text{eff}} = 3.56 \pm 0.23$ [14]. In comparison the 2013 Plank measurement of the cosmic microwave background gives $N_{\text{eff}} = 3.30 \pm 0.27$ [15], and the Standard Model prediction is $N_{\text{eff}} = 3.04$ [16].

## 4. Neutron Lifetime Experiments

Experiments that measure the neutron lifetime fall into two broad categories: *beam experiments* that measure the specific activity of a neutron beam by simultaneously counting the decay products and the number of neutrons in the decay region; and *ultracold neutron storage experiments* where ultracold neutrons are stored and the number that remain after a known decay period is counted. We now present an overview of these two methods. A more detailed and historical review can be found in [17].

### 4.1. Beam Experiments

The beam method is the oldest, used in the first serious measurement of the neutron lifetime by Robson at the Chalk River pile reactor in 1950 [1]. A beam of slow (thermal velocity or less) neutrons with density $\rho_n$ passes through a decay volume $V$. The neutron decay rate $\dot{N}$ in the decay volume is given by the exponential decay differential equation

$$\dot{N} = \frac{dN}{dt} = -\frac{N}{\tau_n} = -\frac{\rho_n V}{\tau_n}. \tag{10}$$

We can further write

$$\rho_n = \int \frac{\phi(v)}{v} dv \tag{11}$$

where $\phi(v)$ is the neutron spectral flux (assumed here for simplicity to be spatially uniform), $v$ is the neutron velocity, and the integral is taken over the neutron beam velocity spectrum. Also

$$V = A_{\text{beam}} L_{\text{det}} \tag{12}$$

where $A_{\text{beam}}$ is the beam area and $L_{\text{det}}$ is the length of the neutron decay volume. Combining Equations (10)–(12) we have

$$\tau_n = \frac{A_{\text{beam}} L_{\text{det}}}{\dot{N}} \int \frac{\phi(v)}{v} dv \tag{13}$$

After exiting the decay volume the neutron beam passes through a thin absorbing foil. The neutron absorption cross section in the foil is inversely proportional to neutron velocity, the "$1/v$ law" [18]. It is not an exact law, but it holds to excellent approximation with relative error less than $10^{-4}$ in many materials, the exceptions being very strong neutron absorbers and isotopes with low energy neutron capture resonances. For a "$1/v$" absorber the absorption cross section is

$$\sigma_{\text{abs}}(v) = \frac{\sigma_{\text{th}} v_{\text{th}}}{v}, \tag{14}$$

where $\sigma_{\text{th}}$ is the thermal neutron absorption cross section of the foil material at the reference thermal neutron velocity $v_{\text{th}} \equiv 2200$ m/s. Integrating over the velocity spectrum of the neutron beam, the measured rate of reaction products from neutron absorption in the foil is then

$$R_n = (\varepsilon_n \rho_{\text{foil}} \sigma_{\text{th}}) A_{\text{beam}} \int \frac{\phi(v)}{v} dv = \varepsilon_0 A_{\text{beam}} \int \frac{\phi(v)}{v} dv. \tag{15}$$

Here $\varepsilon_0$ is defined to be the probability of detecting a reference thermal neutron (2200 m/s) incident on the foil, equal to the product of $\sigma_{\text{th}}$, the areal density $\rho_{\text{foil}}$ (atoms per unit area) of the foil, and the detection efficiency $\varepsilon_n$ for reaction products from the foil.

Neutron decay products, electrons and/or protons, from decays inside the decay volume are measured with efficiency $\varepsilon_p$, giving a measured neutron decay rate

$$R_p = \varepsilon_p \dot{N}. \tag{16}$$

Combining Equations (13), (15) and (16) we obtain an expression for $\tau_n$ in terms of quantities measured in an experiment

$$\tau_n = \frac{R_n \varepsilon_p L_{\text{det}}}{R_p \varepsilon_0 v_{\text{th}}}. \tag{17}$$

Note that the integrals in Equations (13) and (15) are identical so they cancel in the ratio, Equation (17). The probability for a neutron to decay inside the decay volume is proportional to $1/v$, its time traversing it. This is precisely compensated by the $1/v$ probability for a neutron to be counted by the absorbing foil. Therefore in the beam method the neutron lifetime is relatively insensitive to neutron velocity and a broad spectrum high flux neutron beam can be used.

Free neutron beta decay was first observed in 1950 by Snell, Pleasonton, and McCord [19] at the Oak Ridge Graphite Reactor and independently by Robson [20] at the Chalk River reactor in Canada. Both estimated the neutron lifetime to be in the range 10–30 min, consistent with expectation at the time and with the current value. Robson continued to make the first earnest neutron lifetime measurement, using what is now called the beam method. A collimated beam of thermal neutrons from the reactor passed through a chamber in which decay electrons and protons were detected in coincidence, and then through a thin manganese foil. The foil was removed and placed in a counting chamber to measure the neutron induced radioactivity and hence the neutron count rate $R_n$. A neutron beam will generate a large prompt gamma ray background in its vicinity due to neutron capture in neighboring materials, and this gamma ray rate is typically much larger than the neutron decay rate, so background reduction is a challenge for the beam method. Robson's electron-proton coincidence method enabled a significant reduction in detector backgrounds but it had the disadvantage of causing the efficiency $\varepsilon_p$ to depend strongly on decay position, creating a large systematic uncertainty. He reported a half-life of $12.8 \pm 2.5$ min, or $\tau_n = 1110 \pm 220$ s [1].

A marked improvement in precision of the neutron lifetime was achieved by Christensen, et al. working at the Risö reactor in Denmark [21] in 1971. A drawing of the apparatus is shown in Figure 1. Two parallel plastic scintillator paddles placed between the poles of a large electromagnet defined the neutron decay volume. When a neutron decayed, the beta electron was transported in a helical orbit, following the magnetic field, to one of the paddles and detected. If the electron backscattered, it would be transported to the other paddle and detected there as well, with the signals summed. So nearly 100% of the electron energy was registered for each decay. With this method the efficiency $\varepsilon_p$ was very uniform in the decay region, but background gamma radiation was a serious issue. The neutron beam density was measured using a large area $^3$He proportional counter with gold foil activiation intercomparisons. The result was $\tau_n = 918 \pm 14$ s.

The best precision in beam neutron lifetime measurements has been achieved over a span of four decades by the Sussex-ILL-NIST series of experiments [22–26]. A well-collimated cold neutron beam passes through a quasi-Penning trap; see Figure 2. The trap consists of 16 annular electrode segments. In the trapping state, the first three segments (the "door") are held at +800 V, a variable number (3–10) of trap segments are held at ground, and the following three segments (the "mirror") are also at +800 volts. When a neutron decays inside the central, grounded region of the trap, the recoil proton is trapped radially by the 4.6 T magnetic field and axially by the electrostatic potential of the door and mirror. After some period of time in the trapping state, typically 10 ms, the door electrodes are lowered to ground and a small ramped potential is applied to the trap electrodes to flush trapped protons through the door and toward a silicon proton detector via a 9.5° bend in the magnetic field. The detector is held at a $-30$ kV (typically) potential so that protons will be sufficiently energetic to penetrate the detector dead layer and produce a countable electronic pulse. At the end of the counting

period, about 80 μs, trap electrodes are returned to the trapping configuration and the cycle repeats. After exiting the trap the neutron beam passes through a very thin deposit of $^6$LiF on a silicon substrate. A set of four surface barrier detectors with very well characterized detection solid angle and efficiency counts the alpha and triton products from $^6$Li$(n, \alpha)^3$H reactions to determine $R_n$.

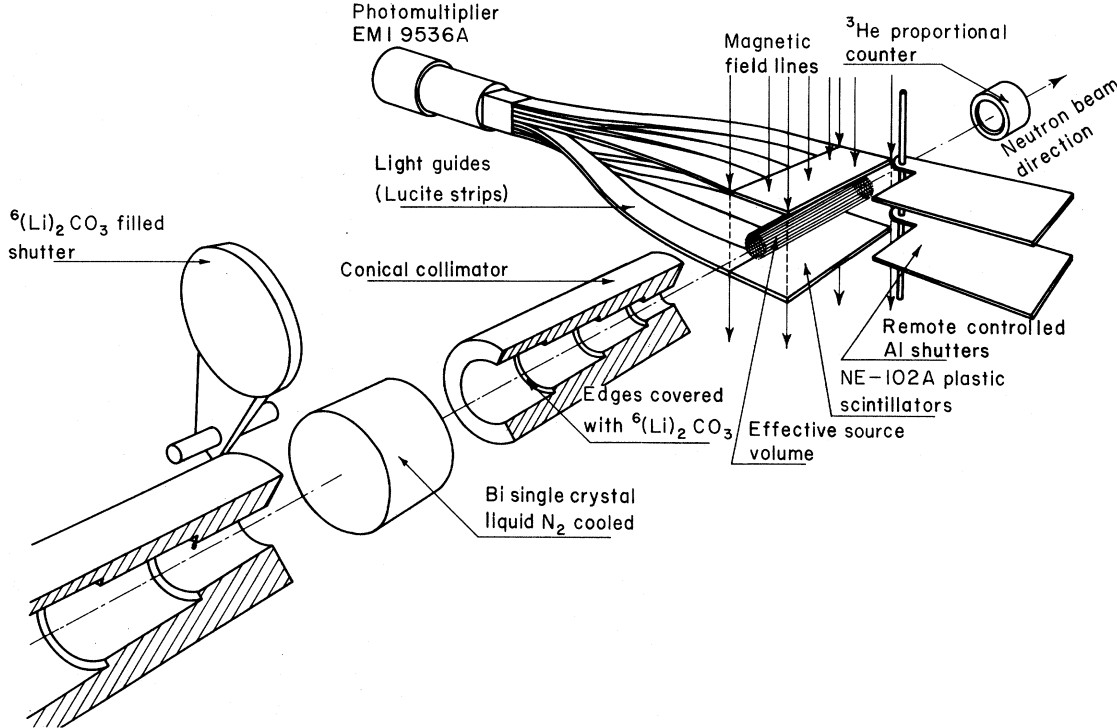

**Figure 1.** The Risö neutron lifetime experiment [21]. The decay volume is defined by two parallel plastic scintilator paddles within a uniform magnetic field.

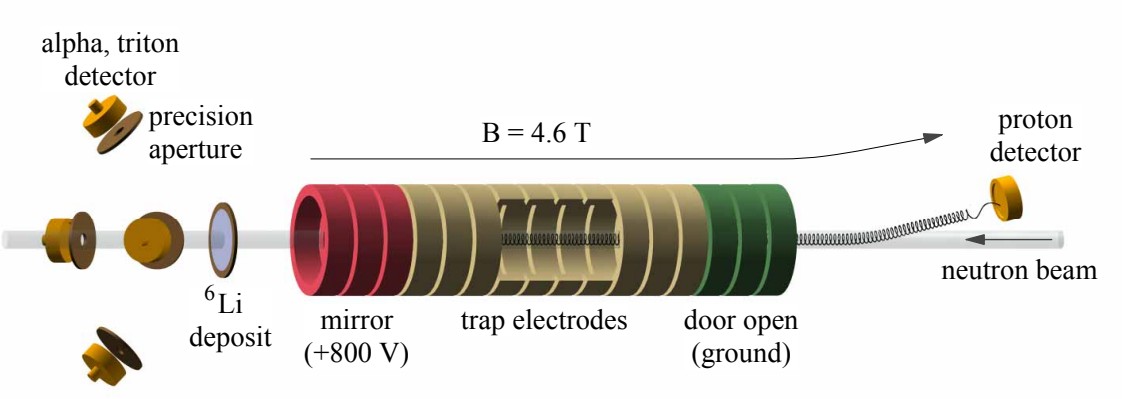

**Figure 2.** Scheme of the beam neutron lifetime experiment using the Sussex-ILL-NIST method [25]. The neutron beam passes through a segmented quasi-Penning trap. Decay protons are trapped by the elevated door and mirror electrode potentials and counted periodically by lowering the door to ground as shown. Neutrons are counted by detecting the alphas and tritons from the $(n, \alpha)$ reaction in a thin $^6$LiF deposit.

The proton trapping efficiency is effectively 100% for protons born in the central ground region of the trap where they have insufficient energy to escape. But protons can also be born in the region of the door and mirror electrodes where their electrostatic potential energy is elevated. Such protons can escape with a probability that depends on their initial position and momentum. This is why the trap is segmented. The decay region length can be written $L_{det} = nl + L_{end}$, where $n$ is the number of grounded trap electrodes, $l$ is the known electrode length, and $L_{end}$ is the *effective* length of the door and mirror trapping regions, which incorporates both the physical lengths of the door and mirror and the probability that a proton born within them will be trapped. The value of $L_{end}$ is unknown, but if the magnetic field, trap geometry, and neutron beam are sufficiently symmetric it is constant for all trap lengths i.e., with $n = 3$–10. Equation (17) can then be written

$$\frac{R_p}{R_n} = \tau_n^{-1} \left( \frac{\varepsilon_p}{v_{th}\, \varepsilon_0} \right) (nl + L_{end}).$$ (18)

The experimentally measured ratio $R_p/R_n$ vs. $n$ is fit to a straight line and $\tau_n$ is found from the slope, so it is independent of $L_{end}$. An example of such a fit is shown in Figure 3 (top).

The proton detection efficiency $\varepsilon_p$ is greater than 98%. It differs from unity due to the possibility of Rutherford scattering or energy straggling in the detector dead layer such that the proton does not deposit sufficient energy in the active volume to produce a signal above threshold. If the proton backscatters after depositing little energy, it has some chance to be reflected by the electrostatic field back to the active region of the detector and subsequently detected, and that probability is difficult to estimate accurately. The most effective strategy is to make lifetime measurements with different detectors, with dead layer thickness and composition, and varying acceleration potential. For each condition the loss due to backscatter and straggling is computed by Monte Carlo. The lifetimes are then extrapolated to the zero backscatter point, where in principle $\varepsilon_p = 1$. This extrapolation is shown in Figure 3 (bottom).

The limiting systematic uncertainty in the 2005 NIST beam neutron lifetime experiment [25] was associated with $\varepsilon_0$, computed from

$$\varepsilon_0 = \frac{2N_A\sigma_0}{4\pi A} \int\int \Omega(x,y)\rho(x,y)\phi(x,y)dxdy$$ (19)

where $N_A$ is the Avogadro constant and $A$ is the atomic weight of $^6$Li. $\Omega(x,y)$ is the detector solid angle as a function of $x, y$, the planar coordinates on the foil, computed from the precisely measured geometry of the detector apertures. The function $\rho(x,y)$ is the areal density of $^6$Li in the foil measured separately in a program of isotope dilution mass spectroscopy and microscopic measurements of the deposit; and $\phi(x,y)$ is the neutron beam spatial distribution funtion measured using neutron imaging techniques. The tabulated $^6$Li $(n,\alpha)$ cross section $\sigma_0$ was determined from an $R$-matrix evaluation [27] with an uncertainty of 0.14%. The uncertainties in $\rho(x,y)$ and $\sigma_0$ alone accounted for 2.5 s of systematic uncertainty in the neutron lifetime, with little hope for improvement absent a new strategy. This problem was recognized long ago [28] and led to a program to develop an absolute neutron flux measurement capability that could be used to independently calibrate the lifetime experiment neutron counter and eliminate these sources of error. In 2011 an alpha-gamma spectrometer developed at NIST succeeded and the calibration was completed with 0.06% relative uncertainty [26]. The new calibration was applied to the 2005 data and the updated value was slightly higher but consistent. With the systematic bottleneck due to the neutron counter efficiency removed, incremental improvements can be made to the existing NIST apparatus and the experiment repeated. A new run is in progress as of this writing with a goal of 1.0 s precision.

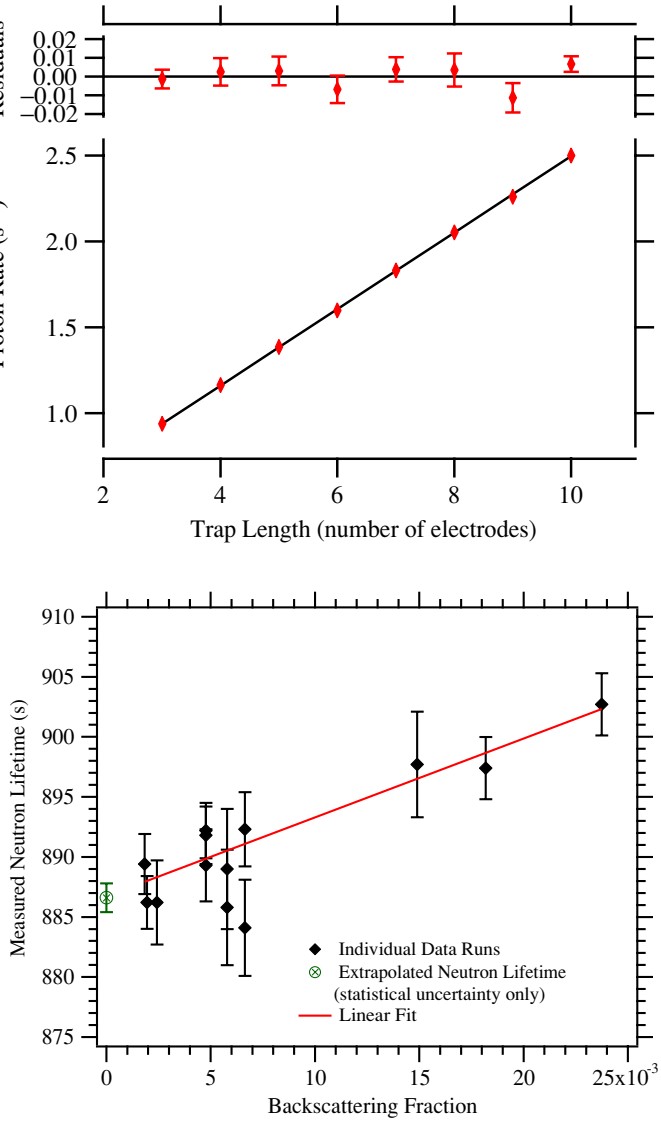

**Figure 3.** (**Top**): Data from the 2005 NIST beam neutron lifetime experiment [25], showing the proton count rate vs. trap length. (**Bottom**): Measured neutron lifetimes using different detectors and acceleration potentials vs. backscatter probability. The corrected neutron lifetime is found by linear extrapolation to zero backscatter.

A next-generation version of the Sussex-ILL-NIST method, called BL3 ("Beam Lifetime 3"), is currently under development. It will employ a new, much larger magnet and trap to accomodate a larger neutron beam for much higher proton counting rate. A segmented, large area proton detector will be used. A number of systematic improvements in proton counting, mitigation of proton backscatter, and magnet/trap uniformity are planned, with an ultimate uncertainty goal of 0.1 s in the neutron lifetime.

A new beam neutron lifetime experiment [29–31] is underway at the Japanese Proton Accelerator Research Complex (J-PARC). A pulsed cold neutron beam passes through a gas time projection chamber (TPC) containing a dilute admixture (100 mPa) of $^3$He. The TPC simultaneously detects and counts neutron decay betas and measures neutron density via $^3$He($n,p$)T reactions. The neutron counter is "thin" in the sense that the $^3$He density is sufficiently low that the $1/v$ law applies. A spin flip chopper produces very short neutron bunches; a fiducial time cut is then used to eliminate end effects in the TPC.

The experiment first ran in 2016 and data analysis is in progress; a preliminary result of $\tau_n = 896 \pm 16$ s was reported in May 2018 [31]. The ultimate goal is 0.1 s uncertainty in the neutron lifetime.

### 4.2. Ultracold Neutron Storage Experiments

Ultracold neutrons (UCN) are neutrons whose kinetic energy is less than about 100 neV (temperature less than 1 mK, de Broglie wavelength greater than 100 nm). It is an interesting and useful coincidence that at this energy it becomes practical for a neutron to be trapped and stored for long periods using the potential energy of its interaction with solid or liquid materials, a magnetic field, or the Earth's gravitational field. The Fermi effective potential, which originates from coherent scattering of a low energy neutron from atomic nuclei, is in the range 100–300 neV in many materials, so ultracold neutrons can be totally reflected from the interior surface of a material bottle. The potential energy of the neutron's magnetic dipole moment in a magnetic field is about 60 neV per Tesla, so ultracold neutrons can be trapped in a strong inhomogeneous laboratory magnetic field. The *mgh* potential of the neutron at the Earth's surface is 100 neV/m so UCN can be confined vertically by the Earth's gravity in a laboratory-scale apparatus. The prospect of measuring the neutron lifetime using UCN stored in a bottle motivated the development of UCN sources at research reactors in the 1970s. See [32] for a thorough introduction to UCN theory, production, storage, and experimental methods.

The basic idea of the UCN storage method is straightforward. UCN are produced and admitted into a container whose walls have a high potential energy so are totally reflecting for neutrons. If the container is sufficiently tall neutrons can be vertically trapped by gravity. Following some storage time $\Delta t$ the surviving neutrons are extracted and counted. Two (or more) storage times $\Delta t_1$ and $\Delta t_2$ are used, preferably with a range that brackets the neutron decay lifetime. If the only loss mechanism is due to neutron beta decay, the ratio of neutron count rates for the two storage times $N_1/N_2$ gives the neutron lifetime

$$\tau = \frac{\Delta t_2 - \Delta t_1}{\ln\left(\frac{N_1}{N_2}\right)}. \tag{20}$$

In theory this sounds easy but in practice there are always competing neutron loss mechanisms that must be accurately accounted for. These present serious challenges for experiments. The first generation experiments using this method relied on material bottles. One problem is that total reflection of neutrons predicted by the bulk effective potential of the wall material cannot be not fully achieved because impurities at the surfaces cause inelastic upscattering and absorption. The bottle wall temperature is much higher than the 1 mK effective temperature of the stored ultracold neutrons, so inelastic scattering leaves the neutron with a kinetic energy higher than the Fermi effective potential of the wall material and it will quickly escape. Hydrogen has a large incoherent neutron scattering cross section and it is ubiquitous at solid metal surfaces. On reflection the neutron's evanescent wave penetrates the wall surface, so neutron capture within the wall is possible. The experiment's measured neutron storage lifetime $\tau$ in Equation (20) can therefore be written

$$\frac{1}{\tau} = \frac{1}{\tau_n} + \frac{1}{\tau_{\text{inel}}} + \frac{1}{\tau_{\text{cap}}} + \frac{1}{\tau_{\text{other}}}, \tag{21}$$

where $\tau_n$ is the beta decay lifetime, $\tau_{\text{inel}}$ is loss due to inelastic scattering at the walls, $\tau_{\text{cap}}$ is neutron capture at the walls, and $\tau_{\text{other}}$ accounts for any other loss mechanisms such as residual gas interactions or small gaps in the bottle walls that allow UCN to escape. It is important to emphasize that Equation (21) assumes all loss mechanisms occur at constant rate. This is not strictly true in practice as UCN loss rates can depend on the neutron velocity spectrum which may evolve during the neutron storage period.

UCN storage experiments have had to contend with the problem of quasi-trapped neutrons, i.e., neutrons whose mechanical energy exceeds the maximum trap potential but whose trajectories enable them to remain in the container for a relatively long time, thus introducing an additional effective loss mechanism. UCN inside a container with kinetic energy less than the effective potential energy of its walls cannot escape other than by scattering, material absorption, or beta decay. Those with kinetic energy $U_{\mathrm{wall}} + \epsilon$, slightly in excess of the wall potential, will escape, but only if they strike the wall at an angle $\phi < \sqrt{\epsilon/U_{\mathrm{wall}}}$ from normal incidence. For symmetric, smooth containers such as spheres, rectangular boxes, and cylinders there exist stable trajectories for quasi-trapped neutrons that will never satisfy that criterion, and others that will satisfy it only after many reflections. Irregular surfaces produce randomized reflections that tend to reduce quasi-trapped neutrons. Many UCN storage experiments employ some means of "cleaning" the neutron spectrum, i.e., removing the most energetic part by lowering an absorbing material or rotating the container to expel energetic neutrons. Such techniques mitigate, but do not completely eliminate, the problem.

The UCN bottle wall impurity problem was largely solved by applying hydrogen-free coatings to the walls. In particular frozen $D_2O$ [33] and Fomblin oil [34] were shown to be effective and led to much longer UCN storage times. Fomblin is a viscous fluorinated polyether well known for its use in diffusion vacuum pumps. It forms a stable, smooth, renewable surface on glass and has an effective Fermi potential of 107 neV so it is suitable for storing UCN.

The MAMBO UCN storage bottle experiments [35–37] were based at the UCN turbine source PF-2 at the Institut Laue-Langevin (ILL) in France. A sketch of the first MAMBO apparatus is shown in Figure 4 (top). If the trapped neutrons are monoenergetic and their trajectories fill the allowed phase space, the wall loss rate can be calculated by kinetic theory

$$\frac{1}{\tau_{\mathrm{wall}}} = \frac{\mu v}{\lambda}, \tag{22}$$

with $\mu$ the loss probability per bounce, $v$ the neutron velocity, and $\lambda$ the mean free path given by $\lambda = 4V/S$, where $V/S$ is the volume to surface ratio in the bottle. The rear wall of the bottle was moved by a piston in order to vary the surface to volume ratio and the movable wall contained sinusoidal corregations to randomize the trajectories of reflected neutrons. An issue arises from the fact that the neutrons are not monoenergetic and $\mu$ is velocity dependent so $\tau_{\mathrm{wall}}$ varies in a complicated way with time. At the beginning of each measurement cycle UCN were admitted into the bottle under precisely the same conditions so the initial neutron velocity spectrum immediately after filling is the same. If the storage time $\Delta t$ is chosen to be proportional to $V/S$, then for two sets of measurements with different $V/S$ (different rear wall positions), the average number of wall reflections during $\Delta t_1$ and $\Delta t_2$ will be the same for both sets. Furthermore the neutron velocity spectrum will evolve in the same way for both sets at a rate proportional to $V/S$. Therefore because each set experienced the same number of wall reflections, the data can be linearly extrapolated to zero wall reflection rate, as shown in Figure 4 (bottom).

A second version of the experiment, MAMBO II [36], added a neutron prestorage volume that was filled first. The prestorage volume had a moveable absorbing roof that removed the most energetic neutrons, thus "cleaning" the neutron spectrum prior to admission into the main storage volume in order to reduce the effects of quasi-trapped neutrons.

The Gravitrap series of UCN storage experiments [38–40] were proposed by A. Serebrov and A.V. Strelkov and developed by the Petersburg Nuclear Physics Institute (PNPI) and the Joint Institute for Nuclear Research (JINR) in Russia. Following a shutdown of the PNPI UCN source the program relocated to the ILL. A drawing of the first Gravitrap bottle is shown in Figure 5. The neutron bottle was a 75-cm diameter spherical container coated with beryllium and cooled to below 15 K to reduce upscattering at the walls. It had a small hole at the top and could be rotated about a horizontal axis. After filling the bottle with UCN from the source, it was rotated by a reproducable angle to allow neutrons whose energy exceeded the gravitational potential at the height of the hole to escape.

The bottle was then rotated back to return the hole to the top and the neutron storage lifetime was measured. The purpose of the rotation was twofold: (1) to vary the initial neutron velocity spectrum, and hence the wall loss rate, so that an extrapolation to zero loss rate (zero neutron velocity) could be made in the data analysis; and (2) to clean the neutron spectrum and thereby reduce quasi-trapped neutrons. The experiment was repeated using a solid oxygen wall coating and achieved an improved result [39].

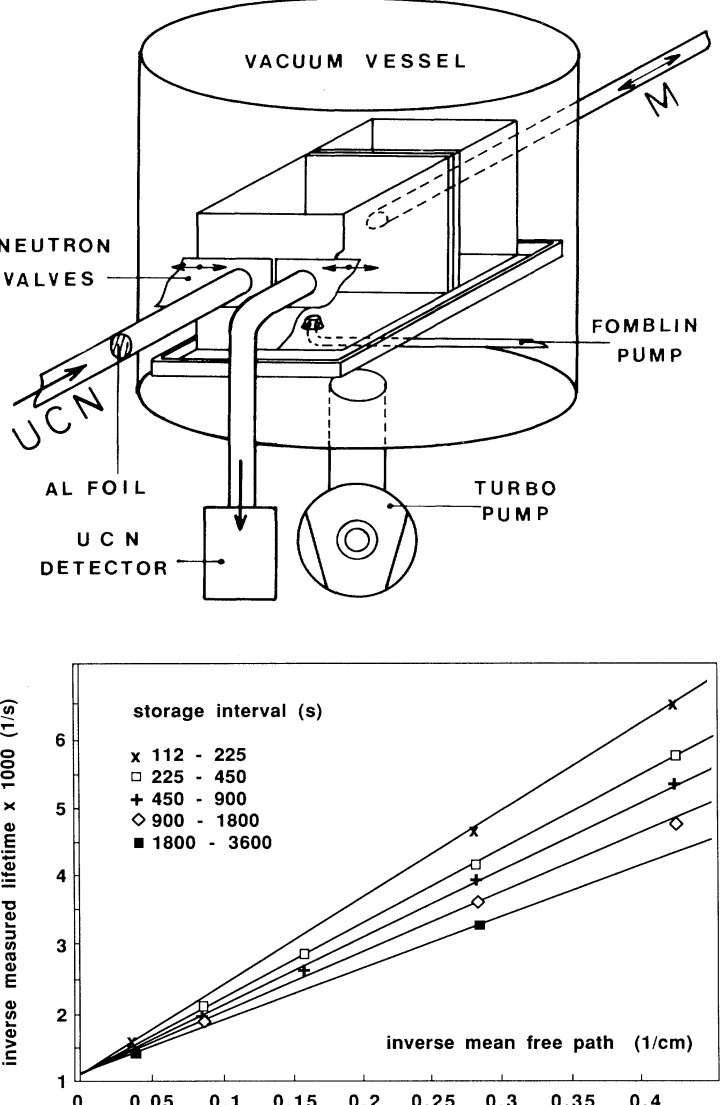

**Figure 4.** (**Top**): The ultracold neutron bottle used in the first MAMBO experiment [35]. Neutrons entered through the straight tube in front during the fill phase and exited through the bent tube during the counting phase. The rear wall could be moved using rod M to change the volume to surface ratio of the bottle and thereby control the wall loss rate. (**Bottom**): Data from the first MAMBO experiment; the measured inverse bottle lifetimes as a function of bottle inverse mean free path, for different storage intervals corresponding to different positions of the rear wall. The data extrapolate to a single point at zero inverse mean free path (infinite volume to surface ratio), where the wall loss rate is zero in principle, giving the neutron beta decay lifetime.

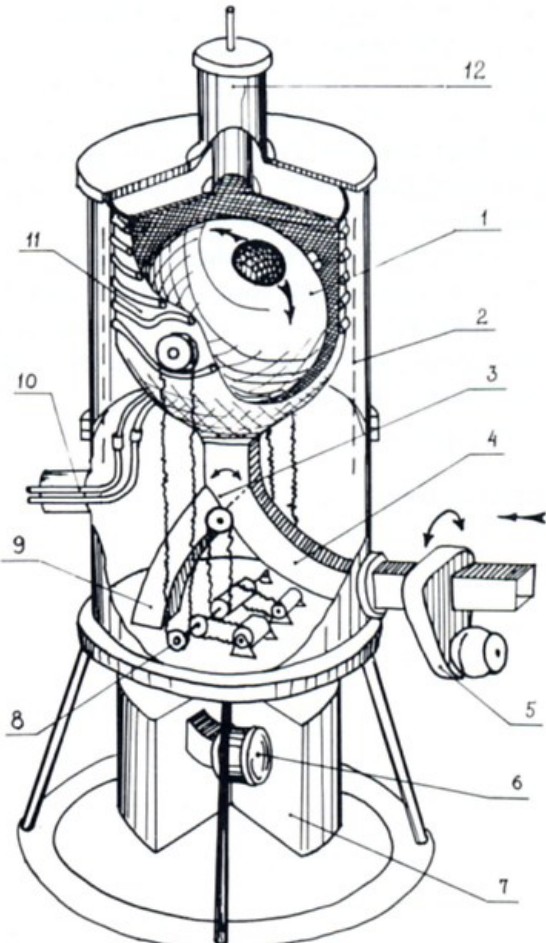

**Figure 5.** The original Gravitrap neutron lifetime bottle apparatus [38]: (1) UCN storage bottle; (2) liquid nitrogen screen; (3) neutron distribution valve; (4) neutron guide; (5) neutron filling valve; (6) $^3$He neutron counter; (7) radiation shielding; (8) rotational control linkage; (9) neutron guide; (10) cryogen lines; (11) cryostat; (12) frozen oxygen system.

The Gravitrap experiment was subsequently modified to allow a coating of fluorinated polyether similar to Fomblin that can be evaporated onto a surface and forms a stable coating at cryogenic temperatures, in this case 113 K [40]. With this coating the inelastic scattering probability per bounce was observed to be about $2 \times 10^{-6}$, a factor of ten smaller than for liquid Fomblin oil. An additional improvement was the use of two nested cylinders, a narrow cylinder and a larger quasi-spherical bottle, both mounted on the same rotatable axis, so the neutron velocity spectrum and the volume to surface ratio could be varied independently to modulate the wall loss rate. With this new cryogenic wall coating the full range of extrapolation from the measured storage times to the zero wall loss point was only 15 s, much smaller than in all previous bottle experiments, and consistent with the lower wall loss rate. To test the integrity of the coating, additional measurements were made using a titanium bottle coated with the cryogenic oil. Titanium has a negative Fermi potential, so any UCN that can reach the titanium wall via a small hole or defect in the crygenic coating could not be reflected and would be immediately lost. With this bottle the neutron storage time was measured to be 869 s, just slightly shorter than in the main experiment where the bottles had beryllium (large positive Fermi potential so a good neutron reflector) under the cryogenic oil. This demonstrated that the fraction of the bottle surface area that was not effectively coated with the oil was less than $10^{-6}$. The experiment obtained the result $\tau_n = 878.5 \pm 0.7_{\text{stat}} \pm 0.3_{\text{sys}}$ s [40,41], about 7 s lower and in significant disagreement

with the world average [42] at that time, initiating the "neutron lifetime discrepancy" that persists to the present; see the discussion in Section 5.

A next generation, larger and improved, version of the Gravitrap was constructed and run recently at the ILL. The main UCN bottle was a copper half-cylinder mounted on a horizontal shaft as before. It was coated with a fluorinated polyether grease operated at 80 K to minimize wall losses. The bottle was about a factor of five larger in volume than the original spherical Gravitrap bottle giving a significant increase in count rate. A second, smaller copper half-cylinder could be inserted to reduce the volume/surface ratio. The experiment was operated in a similar manner as the previous, rotating the bottle prior to measurement to both clean and vary the initial neutron velocity spectrum, and an extrapolation via a Monte Carlo calculation to zero wall losses. The first result was presented in 2018: $\tau_n = 881.5 \pm 0.7_{\text{stat}} \pm 0.6_{\text{sys}}$ s [43]. Note that this is 3.0 s ($2.5\sigma$) higher than the 2005 Gravitrap result [40].

An emergent strategy to prevent wall losses in UCN storage experiments has been the development of "magnetic bottles". If the container wall is strongly magnetized then an incident neutron will feel a force $\vec{F} = (\vec{\mu} \cdot \nabla)\vec{B}$ from the interaction of the neutron's magnetic dipole moment $\vec{\mu}$ with the magnetic field gradient. The force will either be attractive (high field seeking) or repulsive (low field seeking) depending on the neutron spin state. Low field seeking neutrons are reflected by the gradient, avoiding a material interaction that can lead to loss by upscattering or absorption. For this to work stored UCN must be 100% polarized. Loss of UCN due to depolarization is a concern; a sufficient holding magnetic field must be applied to the entire volume of the container, and zero points in the field within it must be assiduously avoided to prevent Majorana spin flips.

A magnetic UCN storage neutron lifetime experiment was recently run at the ILL [44]. The container consisted of a 20-pole permanent magnet array on the sides with a solenoid to close the bottom. The container was loaded with UCN by means of an adiabatic mechanical lift. When the solenoid was energized, one neutron spin state was trapped magnetically, and gravitationally at the top. Trapped neutrons could not reach the material walls of the container, but it was nonetheless coated with Fomblin oil so that the high field seeking neutrons would reflect from the walls and escape through the hole in the bottom to be counted. This served as an in situ monitor for depolarized or quasi-trapped neutrons leaking from the container. After a variable storage period of up to 2200 s the solenoid was switched off and the trapped neutrons exited through the bottom.

The UCN$\tau$ experiment [45,46], based at Los Alamos National Laboratory, is the most sophisticated magnetic UCN storage neutron lifetime experiment to date. The storage vessel has an asymmetric "bathtub" shape, a composite of two toroidal sections open at the top, shown in Figure 6. The walls of the vessel are lined with a Halbach array (see Figure 7) of permanent magnets to produce a periodic (in two spatial dimensions) magnetic field gradient. This has a similar effect as the corregated material wall used in the MAMBO experiment; neutron reflection angles from the wall were randomized to suppress quasi-stable neutron orbits. UCN were vertically trapped by gravity. Prior to filling the vessel, UCN passed through a neutron spin polarizer and adiabatic spin flipper to be nearly 100% low field seeking inside the vessel. Any residual wrong spin neutrons were rapidly lost at the walls. A holding field of 6.8 mT inside the vessel prevented neutron depolarization. A horizontal polyethylene sheet (the "cleaner") was lowered for a time sufficient to remove the most energetic neutrons from the spectrum and then raised up, after which the storage period began. At the end of the storage period an in situ neutron detector was lowered into the vessel. It was a large plastic paddle ("dagger") coated with $^{10}$B to absorb neutrons via the $(n, \alpha)$ reaction and ZnS:Ag to produce scintillation light from ionization by the $\alpha$ particles. The scintillation light was transported above to a pair of photomultiplier tubes via wavelength-shifting optical fibers. The novel use of an in situ neutron detector avoided systematic problems from inefficiencies in transporting UCN to an external detector after storage. An important success of the UCN$\tau$ experiment was the lack of (known) competing UCN loss mechanisms compared to previous storage experiments. The Halbach array ensured that upscattering and absorption at the walls would be absent; no extrapolation to zero wall loss was needed. The largest correction made to the measured storage times was due to slow neutron

heating from wall vibrations, which would couple mechanically to the UCN via the Halbach array field. This was a 0.24 s correction estimated from the statistical limit on neutrons detected above the cleaner height.

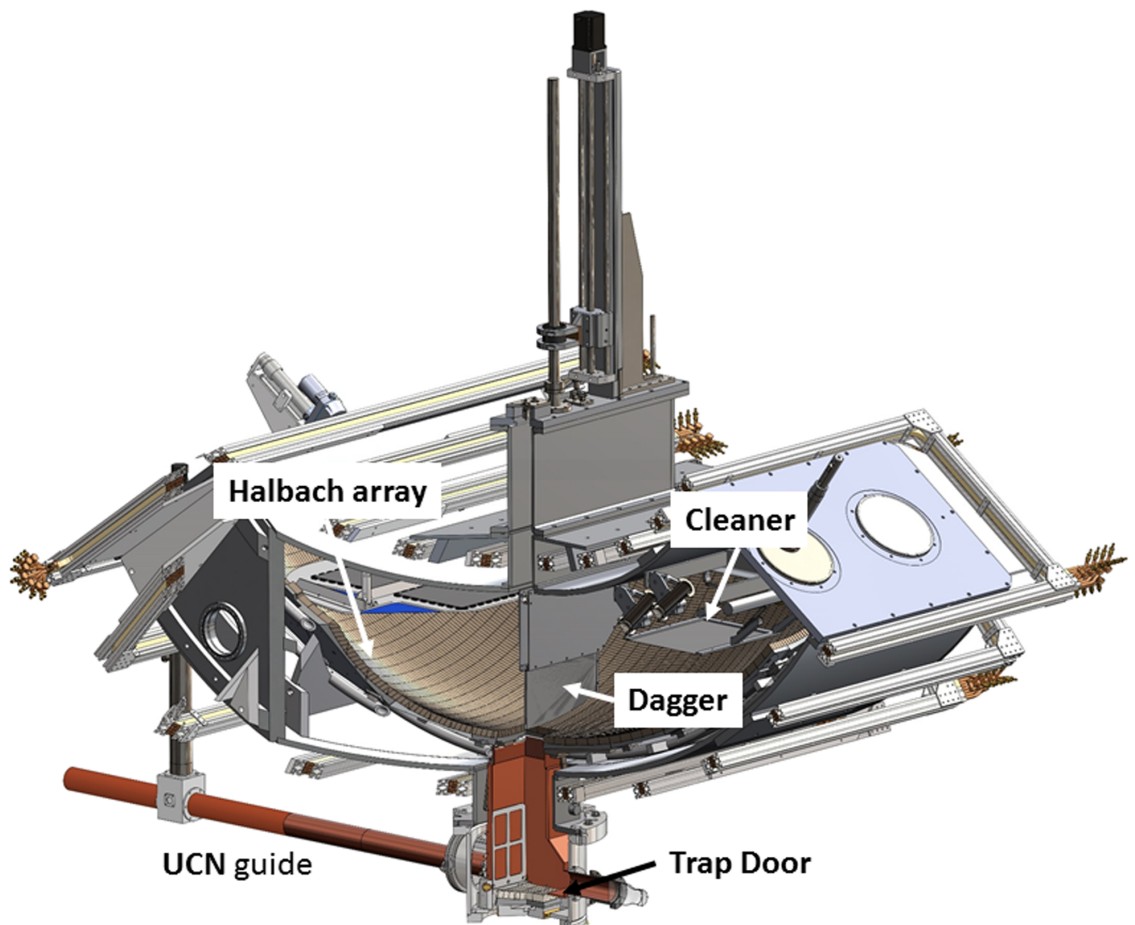

**Figure 6.** A rendering of the UCNτ apparatus [45,46] showing the Halbach array installed on the walls of the UCN storage vessel, the polyethylene sheet used to clean the initial neutron velocity spectrum, and the insertable in situ neutron scintillation detector (dagger). To fill the vessel, polarized neutrons are admitted via the trap door at the bottom.

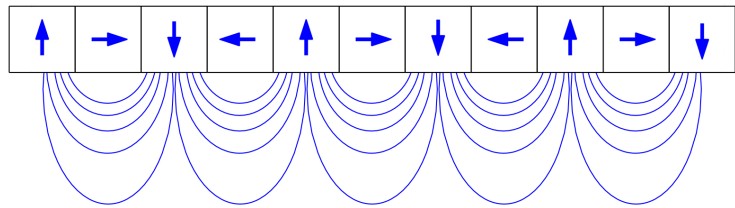

**Figure 7.** A Halbach array of permanent magnets. The arrows indicate the north poles, each rotated 90° from its neighbors. This scheme produces a periodic magnetic field gradient at the surface as shown.

## 5. Discussion and Future Prospects

Figure 8 shows a summary of all neutron lifetime experimental measurements with a reported total uncertainty of less than 10 s. These span the past thirty years. Solid circles indicate the beam method and open squares are the UCN storage method. Dark points are current and are included in the averages (shaded bars). Gray points have been either withdrawn or superceded by later work. We see that prior to 2005 there was good overall agreement between the methods with an average value of $\tau_n = 885.7 \pm 0.8$ s [42]. Then in 2005 the Gravitrap group reported their significantly lower result $\tau_n = 878.5 \pm 0.8$ s [40]. Not surprisingly this was greeted with initial skepticism by the broad neutron physics community, but the fact that the wall loss rate was significantly less than previous UCN storage experiments, and therefore the extrapolation to zero wall losses was much smaller, was compelling. The 2006 Particle Data Group Review of Particle Physics stated "The most recent result, that of [40], is so far from other results that it makes no sense to include it in the average. It is up to workers in this field to resolve this issue. Until this major disagreement is understood our present average of $885.7 \pm 0.8$ s must be suspect" [47].

These events were followed by a general trend of lower neutron lifetime results from UCN storage experiments. In 2010 the MAMBO II experiment [36] found a result slightly higher but consistent with Gravitrap 2005. Shortly thereafter Steyerl et al. published a reanalysis of the MAMBO I data including a new, more sophisticated, estimate of quasielastic scattering effects [37]. This is indicated by the dashed line A in Figure 8. In 2012 the group Arzumanov et al. [48] reported a reanalysis of their previous 2000 result [49] reevaluating a number of systematic effects, in particular the differences between detection efficiencies for neutrons in the two geometries used. The updated value was lower than the original by 3.8 s, see dashed line B in Figure 8. A subsequent experiment by the same group measured $880.2 \pm 1.2$ s [50]. Three new UCN storage measurements were published in 2018 [43,44,46], all clustered around 880 s. A weighted average of the nine current UCN storage lifetime measurements shown in Figure 8 gives $\tau_n = 879.45 \pm 0.39$ s with $\chi^2_{\mathrm{red}} = 2.22$. Expanding the uncertainty by $\sqrt{\chi^2_{\mathrm{red}}}$ gives $\tau_n = 879.37 \pm 0.58$ s.

The 2005 NIST beam neutron lifetime measurement was updated in 2013 using the alpha-gamma calibration as discussed in Section 4.1, raising the result by 1.4 s (the dashed line C in Figure 8) and worsening the disagreement with the UCN storage average. A weighted average of the three current beam lifetime measurments shown in Figure 8 gives $\tau_n = 888.1 \pm 2.0$ s.

The beam-UCN storage neutron lifetime difference now stands at $8.7 \pm 2.1$ s $(4.1\sigma)$. This discrepancy has been called the "Neutron Enigma" in the popular literature [51]. Most likely the difference is due to unaccounted systematic effects in one or both of the methods. Numerous new experiments in the U.S., Europe, and Japan are being pursued to investigate this. One strategy for future beam and storage experiments is to significantly increase the neutron decay volume and/or the neutron density to gain statistical power, so that many measurements at the ~1 s level can be made under varied experimental conditions in the hope of finding correlations that may elucidate unknown systematic effects. Another strategy is the beam/storage hybrid experiment, in which a UCN storage experiment is equipped with a system of high-efficiency particle detectors that can also directly count neutron decays. In all future experiments blinded data analysis will be important to prevent possible biases.

We should not neglect the possibility that new physics is responsible for the neutron lifetime discrepancy. For example a hypothetical "dark" decay mode of the neutron would not be seen in beam experiments, which count only neutron decays with charged particles in the final state. UCN storage measurements, on the other hand, measure the total decay lifetime. The existence of such a decay mode would cause the beam lifetime measurements to be higher, as observed, and a "dark" branching ratio of about 1% would explain the discrepancy. Other exotic possibilities to consider are neutron transitions to "mirror" matter and UCN scattering from light dark matter. A number of such possibilities were proposed and/or experimentally tested in 2018 [52–59].

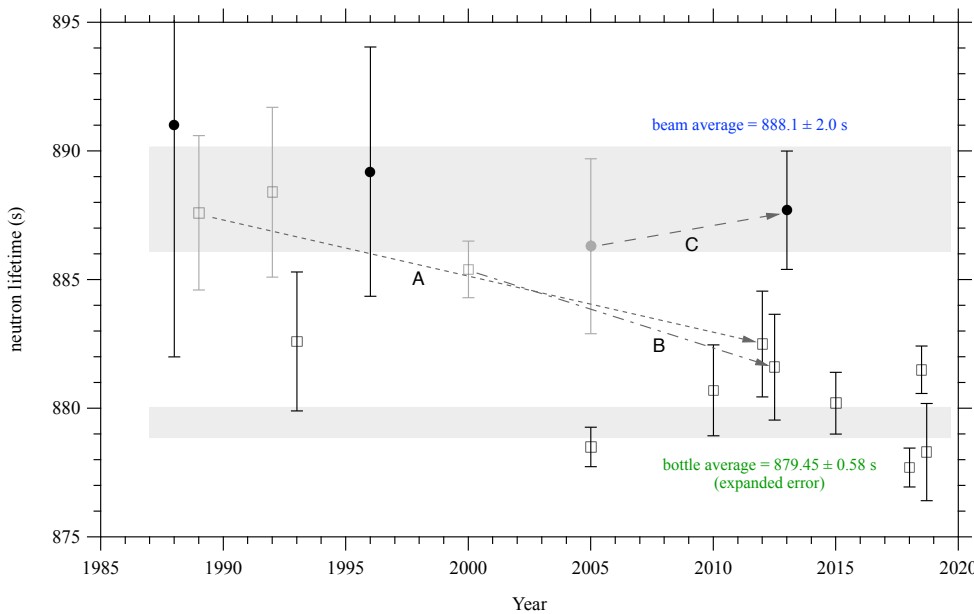

**Figure 8.** A summary of neutron lifetime experimental measurements since 1985. Details of individual measurements can be found in [23,25,26,35–37,39,40,43,44,46,48–50,60,61]. Solid circles are the beam method and open squares are the UCN storage method. Gray points are measurements withdrawn or superceded by later work (old and new indicated by arrows). The shaded bars are weighted averages ±1 standard deviation of uncertainty. The UCN storage uncertainty is expanded (see text). The difference between the beam and storage averages is 8.7 ± 2.1 s (4.1 $\sigma$).

## 6. Conclusions

The value of the neutron lifetime has important implications in particle physics, nuclear physics, and cosmology; and a reliable and precise experimental value is required. While experiments with total uncertainty below 1 s have been reported, in the face of the nearly 9 s discrepancy one cannot say that the neutron lifetime is currently known to the needed precision. New neutron lifetime experiments around the world have been initiated with the aim of elucidating the cause(s) of the discrepancy and improving our knowledge of the neutron lifetime.

**Funding:** The author gratefully acknowledges support from the National Science Foundation grant PHY-1505196.

**Conflicts of Interest:** The author declares no conflict of interest in this work.

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
