# Peer review of "Measurements of the Neutron Lifetime"

_atoms, doi:10.3390/atoms6040070_

Round 1

Reviewer 1 Report

This article provides an excellent and well-balanced summary of the status of precise determinations of a fundamental constant of nature that is notoriously difficult to measure: the lifetime of the free neutron.  It includes a brief summary of the considerable theoretical importance of this quantity that touches on its connection to similar quantities such as decay correlation coefficients, it summarizes past experimental efforts, and it describes the current status of measurements.  The author is highly regarded in the neutron fundamental physics community, and his manuscript shines a bright light on 70 years of work to determine the neutron lifetime.  This article is especially timely as scientists are currently engaged in resolving the apparent beam/bottle lifetime discrepancy that has grown more and more persistent during the last decade.  Given the great potential of this experimental value to test for physics beyond the Standard Model, it is essential that any difference be understood.  I strongly recommend that this well-written article be published so that the greater scientific community can appreciate this fruitful area of fundamental research that efficiently probes the Standard Model using high precision at low energies.

Author Response

I thank the reviewer for a careful reading of the manuscript and recommendation of publication.

Reviewer 2 Report

The article “Measurements of the neutron lifetime” by F.E. Wietfeldt presents an important topic in fundamental particle physics, which attracts particular attention in view of a statistically significant contradiction of results of two series of measurements. Presently, so-called beam experiments show systematically larger values than the results of experiments with trapped neutrons.

The author gives a deep and detailed analysis of physics motivations to measure precisely the neutron lifetime, and then analyzes experiments of these two types.

The description and analysis of beam experiments is correct and precise probably because these experiments are closer to the domain of main expertise of the author.

Some details, however, should be corrected in the section devoted to neutron storage experiments.

-          The cross section of neutron scattering on hydrogen is not equal to 82 bn. In fact, its value depends on its chemical and physical interaction with neighbor atoms. Moreover, it consists of elastic and inelastic parts; the ratio of these partial cross sections and the total cross section depends on temperature. While inelastic cross section is associated with neutron losses, elastic cross section does not contribute to losses;

-          “The use of a Fomblin wall reduced wall losses significantly and enabled the first measured UCN storage times in excess of 700 s” – this is not correct. First neutron lifetime experiments with much longer storage times of UCNs in a trap with heavy-water-coated walls (about 870 s) were performed by the group of Morozov significantly earlier than first MAMBO experiments;

-          Experiments [36-38] were developed by JINR/PNPI collaboration and proposed by A.V. Strelkov/A.P. Serebrov;

-          “The Gravitrap experiment was subsequently upgraded” – might be a too strong statement. In fact, the setup was not upgraded but rather “downgraded” as a fully cryogenic experiment at 4K appeared to be too difficult for only the PNPI part of this (JINR/PNPI) collaboration after its splitting and the relocation of this experiment to ILL;

-          “With this new cryogenic wall coating the wall loss extrapolation from the measured points to the zero point was < 15 s, much smaller than in all previous bottle experiments, and consistent with the lower wall loss rate” – in fact, this is not correct. The storage time was precisely equal to that obtained earlier in [37];

-          “In 2012 and 2015 the group Arzumanov et al. [46,47], based at the Kurchatov Institute in Moscow, reported reanalyses of their previous 2000 result [48]…” – two things are not correct. A) [47] is not a reanalysis but an independent experiment with a new method and a new experimental setup, b) Morozov’s group performed these experiments at ILL;

I recommend to revise the article taking into account the critical remarks given above.

Author Response

I thank the reviewer for a careful reading of the manuscript and thoughtful comments. The reviewer makes a number of good points. My response and manuscript revision for each comment follows.

Reviewer: The cross section of neutron scattering on hydrogen is not equal to 82 bn. In fact, its value depends on its chemical and physical interaction with neighbor atoms. Moreover, it consists of elastic and inelastic parts; the ratio of these partial cross sections and the total cross section depends on temperature. While inelastic cross section is associated with neutron losses, elastic cross section does not contribute to losses.

Response: I was referring to the tabulated scattering cross section computed from the coherent and incoherent scattering lengths, which is indeed 82 b. However this is not the full story as the reviewer points out. The inelastic scattering probability depends on temperature. Also the coherent scattering length of H is negative so it reduces the Fermi potential of the surface. A detailed discussion of these effects is beyond the scope of this brief review and can be found in the cited Ref. 32. The intended point is simply that the presence of H on the bottle surface dramatically increases the probability of UCN upscattering and leads to low storage times.

Revision: Sentence changed to "Hydrogen has a large incoherent neutron scattering cross section and it is ubiquitous at solid metal surfaces."

Reviewer: “The use of a Fomblin wall reduced wall losses significantly and enabled the first measured UCN storage times in excess of 700 s” – this is not correct. First neutron lifetime experiments with much longer storage times of UCNs in a trap with heavy-water-coated walls (about 870 s) were performed by the group of Morozov significantly earlier than first MAMBO experiments.

Response: The reviewer is quite right. I confess I was unaware until now of the success using frozen D2O in the former Soviet Union in the 1980's. Unfortunately this is not well known in the U.S. neutron physics community. It makes me wonder why this idea never caught on. Apparently a precision UCN bottle lifetime experiment using frozen D2O was never completed. Most subsequent experiments used some variation of Fomblin, including those of Morozov's group at the ILL. 

Revision: The paragraph is rewritten as: "The UCN bottle wall impurity problem was largely solved by applying hydrogen-free coatings to the walls. In particular frozen D2O [Kosvintsev, et al., 1982] and Fomblin oil [Bates, 1983] were shown to be effective and led to much longer UCN storage times. Fomblin is a viscous fluorinated polyether well known for its use in diffusion vacuum pumps. It forms a stable, smooth, renewable surface on glass and has an effective Fermi potential of 107 neV so it is suitable for storing UCN."

Reviewer: Experiments [36-38] were developed by JINR/PNPI collaboration and proposed by A.V. Strelkov/A.P. Serebrov

Response: Noted and accepted.

Revision: Sentence changed to "The Gravitrap series of UCN storage experiments [36–38] were proposed by A. Serebrov and A.V. Strelkov and developed by the Petersburg Nuclear Physics Institute (PNPI) and the Joint Institute for Nuclear Research (JINR) in Russia."

Reviewer: “The Gravitrap experiment was subsequently upgraded” – might be a too strong statement. In fact, the setup was not upgraded but rather “downgraded” as a fully cryogenic experiment at 4K appeared to be too difficult for only the PNPI part of this (JINR/PNPI) collaboration after its splitting and the relocation of this experiment to ILL.

Response: Accepted.

Revision: "upgraded" replaced by "modified"

Reviewer: "With this new cryogenic wall coating the wall loss extrapolation from the measured points to the zero point was < 15 s, much smaller than in all previous bottle experiments, and consistent with the lower wall loss rate” – in fact, this is not correct. The storage time was precisely equal to that obtained earlier in [37]

Response: I should have been more clear on the point being made. I agree that the maximum storage time obtained in the two experiments was quite similar, about 875 s. But as seen in Fig. 5 of Ref. 37 (Nezvizhevskii, et al.), the range of the extrapolation to zero wall loss was approximately 770-890 s for the Be coated bottle and 830-890 s for the O coated; compared to Fig. 3 in Ref. 38 (Serebrov et al.) where the range was 863-878 s, about 15 s and much smaller than in previous bottle experiments. The neutron lifetime result depends on the full extrapolation and for a precision result it is desirable for that range to be short.

Revision: Sentence changed to "With this new cryogenic wall coating the full range of extrapolation from the measured storage times to the zero wall loss point was only 15 s, much smaller than in all previous bottle experiments, and consistent with the lower wall loss rate."

Reviewer: “In 2012 and 2015 the group Arzumanov et al. [46,47], based at the Kurchatov Institute in Moscow, reported reanalyses of their previous 2000 result [48]…” – two things are not correct. A) [47] is not a reanalysis but an independent experiment with a new method and a new experimental setup, b) Morozov’s group performed these experiments at ILL.

Response: The full sentence in the manuscript was "In 2012 and 2015 the group Arzumanov et al. [46,47], based at the Kurchatov Institute in Moscow, reported reanalyses of their previous 2000 result [48] reevaluating a number of

systematic effects, in particular the differences between detection efficiencies for neutrons in the two geometries used, and added the results of new measurements made at the ILL." What I meant was Ref. 46 was a reanalysis of the 2000 result and Ref. 47 described the new measurements at the ILL. I should have separated these statements to be more clear. Also, my meaning was that Morozov's group was based at the Kurchatov Institute, but indeed the experiments were run at the ILL. 

Revision: Sentence changed to "In 2012 the group Arzumanov et al. [46], reported a reanalysis of their previous 2000 result [48] reevaluating a number of systematic effects, in particular the differences between detection efficiencies for neutrons in the two geometries used. The updated value was lower than the original by 3.8 s, see dashed line B in figure 8. A subsequent experiment by the same group measured 880.2 ± 1.2 s [47]." Also Figure 8 was updated, with Ref. 46 as an additional point; as the original figure was misleading regarding these measurements.

Round 2

Reviewer 2 Report

i am satisfied with the revision of the manuscript and recommend to publish this article in its present form.